# RefGPT: Dialogue Generation
# of GPT, by GPT, and for GPT

**Dongjie Yang**[1,*], **Ruifeng Yuan**[2], **Yuantao Fan**[3], **Yifei Yang**[1,*],
**Zili Wang**[4], **Shusen Wang**[4], **Hai Zhao**[1, 5, *, †]

[1]Shanghai Jiao Tong University, [2] Hong Kong Polytechnic University,
[3] Beijing University of Posts and Telecommunications
[1]{djyang.tony,yifeiyang}@sjtu.edu.cn, [2]ruifeng.yuan@connect.polyu.hk,
[3]yuantaofan@bupt.edu.cn, [4]{ziliwang.do,wssatzju}@gmail.com,
[5]zhaohai@cs.sjtu.edu.cn

## Abstract

Large Language Models (LLMs) have attained the impressive capability to resolve a wide range of NLP tasks by fine-tuning with high-quality instruction data. However, collecting human-written data of high quality, especially multi-turn dialogues, is expensive and unattainable for most people. Though previous studies have used powerful LLMs to generate the dialogues automatically, they all suffer from generating untruthful dialogues because of the model hallucination. Therefore, we propose a method called RefGPT to generate enormous truthful and customized dialogues without worrying about factual errors caused by the model hallucination. RefGPT solves the model hallucination in dialogue generation by restricting the LLMs to leverage the given reference instead of reciting their own knowledge to generate dialogues. Additionally, RefGPT adds detailed controls on every utterance to enable high customization capability, which previous studies have ignored. On the basis of RefGPT, we also propose two high-quality dialogue datasets generated by GPT-4, namely **RefGPT-Fact** and **RefGPT-Code**. RefGPT-Fact is a dataset with 100k multi-turn dialogues based on factual knowledge and RefGPT-Code has 76k multi-turn dialogues covering a wide range of coding scenarios. Our code and datasets are released in https://github.com/mutonix/RefGPT.

## 1 Introduction

General chat models (OpenAI, 2022, 2023; Anthropic, 2023) based on Large Language Models (LLMs) have shown the impressive capability to intention recognition and complete a variety of NLP tasks only via fine-tuning with a small amount of high-quality instruction data (Taori et al., 2023; Chiang et al., 2023; Xu et al., 2023a). However, such high-quality instruction datasets, especially multi-turn dialogues with instructions in vertical domains, requires enormous crowdsource workers with extensive professional knowledge to collect (Ouyang et al., 2022), where the cost is unaffordable for most people.

Previous studies (Peng et al., 2023; Xu et al., 2023b; Ding et al., 2023) have shown the effectiveness of prompting LLMs like GPT-3 (Brown et al., 2020) to generate enormous instructions (single-turn dialogues) or multi-turn dialogues with given human-written instructions or conversation topics as seeds. However, such one-shot or few-shot methods have a common deficiency that they have the risk of generating untruthful and misleading content due to the language model hallucination (OpenAI, 2023; Ji et al., 2023). The reason why the issue of untruthfulness happens is obvious. This is because the quantity of information in seed prompts like human-written instructions or topics is not enough for being converted to the dialogue on a new topic so LLMs have to recite their own knowledge to complete such a new dialogue which may lead to the model hallucination of generating untruthful facts.

Therefore, we introduce RefGPT, a method for generating truthful and customized multi-turn dialogues utilizing the ability of powerful LLMs like GPT-3.5/GPT-4. RefGPT first provides a plain text or a document as the reference and guides the LLMs to leverage the references to generate dialogues. By providing enough information on a new topic as context, LLMs will be prompted not to rely on their own knowledge to generate the dialogues, thus resolving the hallucination issue.

After ensuring the authenticity of the dialogue, we further develop an effective prompting process

---

*Dongjie Yang, Yifei Yang and Hai Zhao are with the Department of Computer Science and Engineering, Shanghai Jiao Tong University, and also with Key Laboratory of Shanghai Education Commission for Intelligent Interaction and Cognitive Engineering, Shanghai Jiao Tong University.

†Corresponding author; This paper was partially supported by Joint Research Project of Yangtze River Delta Science and Technology Innovation Community (No. 2022CSJGG1400).

for RefGPT to guide the LLMs to generate highly controllable dialogues in a specified uniform format which is easy for training. Previous studies (Xu et al., 2023b; Wang et al., 2022) for automatically generating dialogues have very little control over the generated dialogues. For comparison, RefGPT enables LLMs to generate customized multi-turn dialogues with detailed controls on the structure, style, and content, which further gives diversity to the generated dialogues.

Based on the RefGPT, we also propose two new multi-turn dialogue datasets, namely **RefGPT-Fact** and **RefGPT-Code**. Both datasets have English and Chinese versions. RefGPT-Fact and RefGPT-Code consist of 100k and 76k high-quality multi-turn dialogues generated from GPT-4 separately, using the online encyclopedia websites and Github repositories as the references. As long as the content on the online encyclopedia website and Github codes is truthful and reliable, the authenticity of the generated dialogues can be maximally ensured.

Besides the topics in RefGPT-Fact and RefGPT-Code, RefGPT has the potential to generate truthful dialogues on any topics or vertical domains if we give it relevant references. RefGPT enables such people working in a specific domain, e.g., the nuclear industry, to have a high-quality multi-turn dialogues dataset to train a chatbot specializing in such domain using their own knowledge base as the reference.

To sum up, our contributions are stated as follows:

- We propose RefGPT, a method of generating truthful and customized dialogues using powerful LLMs. Given the reliable reference, RefGPT resolves LLM hallucination in dialogue generation to the greatest extent. RefGPT can also enable detailed customization in the structure, style and content of the dialogues.

- With RefGPT, we construct two new multi-turn dialogue datasets using GPT-4, called **RefGPT-Fact** and **RefGPT-Code**. To our best knowledge, RefGPT-Fact is one of the largest multi-turn dialogue datasets based on factual knowledge. And RefGPT-Code is the first and largest synthetic multi-turn dialogue dataset covering nearly all aspects of code scenarios. These have shown the capability of applying RefGPT to generate dialogues in any vertical domain by utilizing corresponding domain-specific documents.

## 2 Related Work

### 2.1 LLM based Dialogue Generation

The high-quality dialogue dataset is considered crucial for the success of current general chat models (Chiang et al., 2023; Köpf et al., 2023). Due to the high cost of human annotation, previous studies have explored the effectiveness of using LLMs for dialogue generation. Self-Instruct (Wang et al., 2022) presents a framework that facilitates the automatic generation of instruction data (single-turn dialogues) by leveraging existing LLMs. The procedure commences with a set of human-written seed tasks and progressively generates new instructions and responses by iteratively bootstrapping both the initial seeds and the newly produced data. Baize (Xu et al., 2023b) generates multi-turn dialogues by leveraging LLMs to engage in a conversation with itself as both user and assistant based on the given seed topics. UltraChat (Ding et al., 2023) follows a similar idea to Baize and adopts two separate LLM APIs in the generation, where one acts as the user and the other acts as the assistant. However, the dialogues produced by these methods are susceptible to hallucination problems and are uncontrollable. Therefore, we present RefGPT as a solution to generate dialogues with truthfulness and customization.

### 2.2 Reference Based Dialogue Generation

QA pair and dialogue generation based on references have also been widely used. One important requirement for these methods is to ensure the truthfulness of the generated QA pairs and dialogues. Previous studies (Ma et al., 2020; Lewis et al., 2021) generate millions of high-quality QA pairs based on corpus documents using special-purpose question generation models. Dialogue inpainting (Dai et al., 2022) extends this line of work to dialogues by transforming passages from Wikipedia into multi-turn dialogues using a masked conversational language model. In this work, we adopt a similar strategy using the LLMs that we take high-quality documents as references to ensure the truthfulness of the generated dialogues.

## 3 Generation Process

In this section, we present the whole process of RefGPT, which generates truthful and customized multi-turn dialogues by prompting the Large Language Models (LLMs) to effectively utilize the

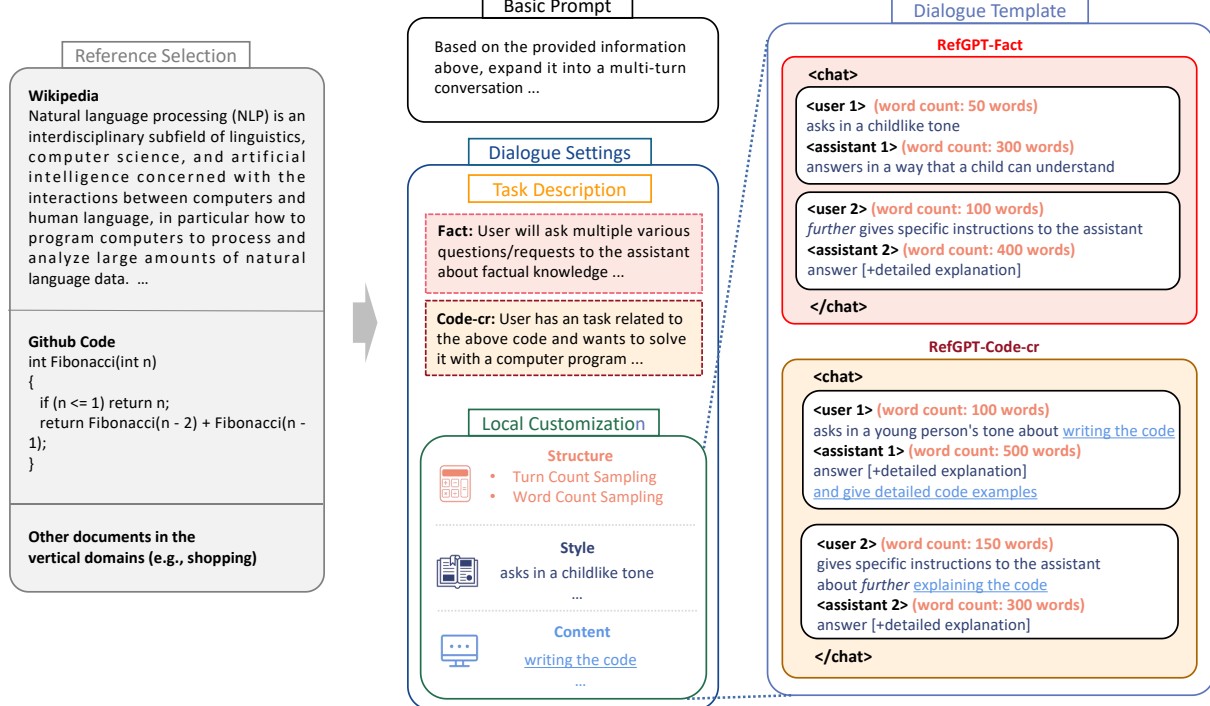

Figure 1: Overview of the whole RefGPT generation process, which mainly consists of three steps: reference selection, basic prompt and dialogue settings.

reference information. As illustrated in Figure 1, the RefGPT process is comprised of three main steps: Reference Selection (pertaining to truthfulness), Basic Prompt, and Dialogue Settings (pertaining to customization).

### 3.1 Reference Selection

RefGPT guides the LLMs to leverage the given external documents or plain texts as references, instead of reciting their own knowledge, to generate truthful dialogues without worrying about hallucination.

The quality of generated dialogues in RefGPT relies on the selection of appropriate references, prioritizing quality and thematic relevance.

A reference in RefGPT can range from a piece of unrefined plain text to a high-quality and dependable document in a specific domain, whose credibility determines the upper limit of the truthfulness of the generated dialogues. On the premise that the reference has contained enough information, it is imperative to opt for high-quality references, such as authoritative knowledge-based websites like Wikipedia.

Furthermore, the chosen reference profoundly influences the thematic direction of the generated dialogues. Consequently, RefGPT exhibits the potential to generate dialogues in diverse domains,

contingent upon the existence of text-based knowledge repositories within those domains. These repositories include a broad spectrum of subjects, including, but not limited to, general domains like factual knowledge with encyclopedias, program codes, and vertical domains like shopping applications or the nuclear industry.

### 3.2 Basic Prompt

To facilitate the generation of multi-turn dialogues that adhere to our basic requirements, we have devised a set of basic prompts:

1. Prompt the LLMs to generate multi-turn dialogues based on the provided reference.

2. Specify the desired language for the dialogue generation. It is preferable for the language of the reference to be consistent with the dialogue to be generated.

3. Instruct the LLMs to reject unreasonable user requests, such as illegal or inappropriate instructions while providing appropriate advice to discourage such actions. This prompt aids in generating dialogues that align with human preferences to a certain extent.

4. LLMs like GPT-3.5-turbo and GPT-4 offer an option of writing a "system" role prompt

to exert precise control over their behaviors in responses. This capability enables customization of the chatbot's identity by providing relevant background information. For instance, in a vertical domain like a shopping app, RefGPT can generate dialogues that conform to the persona of a shopping assistant, even if the reference has no explicit association with shopping (but may have an implicit association).

## 3.3 Dialogue Settings

Rather than generating dialogues uncontrollably, RefGPT uses dialogue settings to convert the reference to a specific dialogue format and customize every utterance, as shown in the middle part of the Figure 1. In dialogue settings, we first specify the task description to tell LLMs how to use the reference. We then customize the structure, style, and content of the dialogue, which can be collectively called local customization.

### 3.3.1 Task Description

We begin by defining the task of dialogue generation concerning the utilization of references, as it relies on the specific aspect of the reference that we aim to initiate the dialogue. For instance, a given piece of program code can lead to multiple scenarios (tasks), such as explaining, creating, or debugging.

### 3.3.2 Local Customization

As per the task description, the local customization specifies the settings regarding the dialogue's structure, style, and content. These settings are then incorporated into a dialogue template for generating the final dialogue.

**Dialogue Structure** To define the dialogue structure, we start the dialogue with the marker `<chat>` and end it with the marker `</chat>`. These two markers specify the range of the whole dialogue. Between the start and the end, we use `<user>` for the user giving instructions and `<assistant>` for the chatbot. A unified output format in a dialogue template avoids most of the weird generations of LLMs and is easier for post-processing. What is more, we will show more merits of using such a format to control the number of turns and length per turn.

**(1) Number of Turns** LLMs like GPT-3.5/GPT-4 often fail with counting the number of the turns of dialogues if we directly require a certain number. But we find that GPT-3.5/GPT-4 are good at following the given format and replacing the placeholders with their own generated content. Therefore, if we want to generate $n$ turns of dialogues, we explicitly give the $n$ `<user>` and `<assistant>` pairs to let LLMs follow the output format. We have also added numerical markers to indicate the $i^{th}$ turn of the dialogue, e.g., `<user i>` and `<assistant i>`, allowing the LLMs to better identify the progress of the current generated turn.

**(2) Length of Utterance** Generating a whole dialogue at one-time, e.g., Self-Instruct (Wang et al., 2022), often leads to much shorter responses than the general chat models like GPT-3.5 do, as shown in Table 1. However, in RefGPT, we can control the lengths of not only the responses of the assistant but also the questions raised by the user at every turn of the dialogue.

We observe that specifying a word count as the prompt is useful for influencing the length of generated utterances. Following the auto-regressive (left-to-right) order, we first illustrate the requirement of word count like `<user>(word count: x words)` or `<assistant>(word count: x words)` before our customization on style and content. Therefore, RefGPT can generate a shorter or much longer question/response depending on the specified word count. Though this prompt can also be used to make the generated utterances longer with other methods like Self-Instruct, generating longer utterances always leads to a more severe hallucination problem. RefGPT filters out the reference whose length is shorter than 80% of the required dialogue length to ensure truthfulness. Thus the LLMs have no necessity of reciting their own knowledge, as the reference length is similar and even longer than the dialogue length.

**Dialogue Style** Staying organized around the same reference, the style of dialogue can vary in the style of asking and answering. For example, a dialogue can start between a user who is a child and an assistant who answers in a way that a child can understand. RefGPT enables this customization for every utterance of `<user>` and `<assistant>` in the dialogues by adding the style requirements before the content customization.

**Dialogue Content** After specifying the style, we can customize the content of each utterance about what to ask and what to answer.

For the task like factual knowledge, the user can be set to ask more about the entity or numbers in the reference. For the task of coding, the user can ask from different perspectives on writing, revising, and using the code and the assistant can choose to give an example or not.

**Dialogue Template** We aggregate the local customizations into a dialogue template to transfer the reference to the dialogue. To enable diversity, we sample different local customization settings for each utterance in the dialogue, as shown in the right-most part in Figure 1. In practice, RefGPT can work well even without style and content pools. These additional settings only need a small amount of manual work for further customization and can be reused to generate diverse dialogues based on different references.

1. For the dialogue structure, we will set the number of turns by weighted sampling. And we sample the word count for both user and assistant in each utterance from a Gaussian distribution.

2. For the dialogue style, we construct a conversational style pool to sample the style settings.

3. For the dialogue content, we construct a content pool according to the task (factual knowledge, code, etc) to sample the content settings.

## 4 RefGPT Dialogue Datasets

In this section, we present two multi-turn dialogue datasets, denoted as **RefGPT-Fact** and **RefGPT-Code**, which are generated utilizing the GPT-4 API in conjunction with RefGPT. More information about these two datasets can be found in Appendix A, and examples are provided in Appendix B.

### 4.1 Dataset Generation Process

**RefGPT-Fact** RefGPT-Fact is a dataset containing 100k multi-turn dialogues about factual knowledge with 50k English and 50k Chinese. The English version uses the English Wikipedia

as the reference and the Chinese version uses the frequently-used Chinese online encyclopedia website, Baidu Baike. We use various dialogue settings mentioned in Sec 3.3 to increase the dialogue diversity.

**RefGPT-Code** RefGPT-Code is a dataset containing 76k multi-turn dialogues about programming with 37k English and 39k Chinese, which has covered most aspects of code usage scenarios and multiple types of programming languages. Both the English version and Chinese version use the public Github dataset on Google BiqQuery with no overlap in these two languages. RefGPT-Code has derived various ways of leveraging the program code as the reference to enable different scenarios. We consider three perspectives of code discussion, code creation and bug fixing in RefGPT-Code.

1. In **RefGPT-Code-ds** about code discussion, we want the LLMs to generate dialogues about asking questions about the given reference code, including explaining, discussing, revising, rewriting, and using the code. After the generation, we will concatenate the reference code as the context to the first question of the user to form the complete version of the dialogue, because we often give the code first before asking questions about it. Thus, the whole dialogue has much longer user utterances, as shown in Table 1.

2. In **RefGPT-Code-cr** about code creation, though we provide the program code as the reference, we assume that the user has an idea/request/trouble/task relevant to the given code but does not know such a code exists, thus he/she wants the assistant to help with writing the code. And the assistant is required to write the code according to the reference code instead of generating a new code to ensure the reliability of the generated code.

3. In **RefGPT-Code-bg** about bug fixing, the user first writes a piece of code with bugs based on the given reference code, which is realized by asking the LLMs to rewrite the code to a buggy one in the first utterance of the user. Then the assistant is required to tell the user where the bugs are and how to fix them according to the reference code. In this scenario, we assume the reference code is reliable and has no bugs.

---

[1]In Table 1, we calculate the number of tokens with the OpenAI tokenizer library of tiktoken in https://github.com/openai/tiktoken. We use cl100k_base in tiktoken to tokenize.

Table 1: Comparsions on different dialogue datasets that contain instructions. **AI** means whether it is generated by AI. **Truthful** indicates whether the truthfulness of the dialogues is guaranteed. **QLen** means the average number of tokens[1] of user utterance. **RLen** means the average number of tokens of assistant utterance. **Turn** means whether the number of dialogue turns can be specified. **Lang** indicates the languages the dataset supports. For a fair comparison, only the English parts are selected in all the datasets.

| Dataset | AI | Truthful | QLen | RLen | Turn | Lang |
|---|---|---|---|---|---|---|
| Dolly (Databricks, 2023) | ✘ | N/A | 16.3 | 78.2 | 1 | en |
| Oasst1 (Köpf et al., 2023) | ✘ | N/A | 28.0 | 169.5 | 1∼5 | multi |
| ShareGPT (Dom Eccleston, 2023) | ✔ | ✘ | 75.6 | 268.8 | 1∼5 | multi |
| Alpaca (Wang et al., 2022) | ✔ | ✘ | 17.2 | 55.3 | 1 | en |
| Baize Quora (Xu et al., 2023b) | ✔ | ✘ | 15.7 | 43.2 | 3∼5 | en |
| UltraChat World (Ding et al., 2023) | ✔ | ✘ | 28.6 | 207.9 | 3∼7 | en |
| RefGPT-Fact | ✔ | ✔ | 28.1 | 269.5 | 3∼4 | en, cn |
| RefGPT-Code-ds | ✔ | ✔ | 281.7 | 374.6 | 3∼4 | en, cn |
| RefGPT-Code-cr | ✔ | ✔ | 36.9 | 395.0 | 3∼4 | en, cn |
| RefGPT-Code-bg | ✔ | ✔ | 155.7 | 380.8 | 2∼4 | en, cn |

## 4.2 Dataset Collection Setup

We use the RefGPT with GPT-4 API to generate these two datasets. The length of every utterance is decided by sampling the Gaussian distribution of $\mathcal{N}(\mu, \sigma)$, where $\mu$ accounts for the average word count (e.g., 300 words) of the utterance and $\sigma$ is the standard variance (e.g., 50 words). The number of turns is decided by weighted sampling, where the weights determine the ratio of dialogues with a specific number of turns in the dataset.

## 4.3 Dataset Statistics

As shown in Table 1, we compare our datasets to other high-quality dialogue datasets. ShareGPT (Dom Eccleston, 2023) collects the dialogues from the real users and ChatGPT, which have much longer user utterances and assistant utterances. If we choose the responses of ChatGPT as a baseline, methods with one API, e.g., Self-Instruct (Wang et al., 2022) and Baize (Xu et al., 2023b), always lead to shorter assistant responses. UltraChat (Ding et al., 2023) with two independent APIs chatting to each other maintains the length of generated responses close to ChatGPT. However, as shown in Table 2, such methods call the model API one utterance at a time with significantly increasing cost and time, as UltraChat has to attach the conversation history multiple times. By contrast, RefGPT generates the whole dialogue with one API call but can adjust the length of generated utterance flexibly according to the requirement.

RefGPT-Fact inherits the diversity of the references like Wikipedia and Baidu Baike. Besides that, RefGPT-Fact has an average response length of 269.5 in English which is very similar to the length of ChatGPT response in ShareGPT.

RefGPT-Code series implements various customizations to be adapted to specific scenarios and have longer user and assistant utterances because we have not only the utterances but also the code attached to the dialogues.

## 5 Experiment

### 5.1 Truthfulness Evaluation

In order to verify the reliability of RefGPT, We evaluate the truthfulness of the RefGPT dataset using both human evaluation for small sample and automatic evaluation with GPT-4 for a large range of verificaiton. For automatic evaluation with GPT-4, though existing methods (Chiang et al., 2023; Liu et al., 2023) have leveraged the GPT-4 to evaluate the performance of other LLMs. However such evaluation is not reliable in factual error checking because GPT-4 has the issue of model hallucination. Inspired by RefGPT, we design a pipeline to evaluate the truthfulness of generated dialogues from our reference datasets, e.g., Wikipedia, by using the GPT-4 to evaluate but with the additional help of reference.

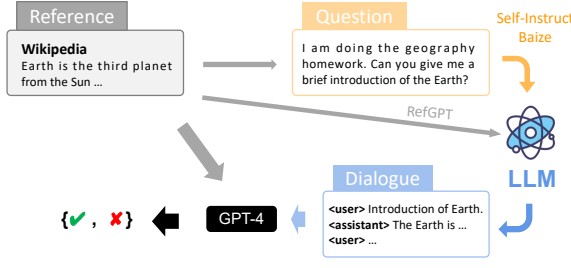

Figure 2: Illustration of the process of truthfulness evaluation.

Table 2: Comparsions on different methods of automatically generating dialogues via LLMs. **Multi-turn** means whether it is a multi-turn dialogue generation. **Human Tru.** and **GPT-4 Tru.** evaluate the truthfulness with accuracy by humans and GPT-4 model. **Len** uses ChatGPT's response length as the standard for long responses. **Turn** means whether the number of dialogue turns can be specified. **Custo.** depends on whether it can control the dialogue structure and content. **Call** is the number of model or model API calls needed for generating an instruction or a $n$-turn dialogue every time.

| Method | Multi-turn | Human Tru. | GPT-4 Tru. | Len | Turn | Custo. | Call |
|---|---|---|---|---|---|---|---|
| Self-Instruct (Wang et al., 2022) | �’ | 54.0 | 50.2 | short | one | limited | 1 |
| Baize Self-Chat (Xu et al., 2023b) | ✔ | 50.0 | 47.2 | short | random | limited | 1 |
| UltraChat (Ding et al., 2023) | ✔ | - | - | long | adjustable | limited | $2n$ |
| RefGPT | ✔ | 98.0 | 97.5 | adjustable | adjustable | highly | 1 |

### 5.1.1 Evaluation Process

We compare RefGPT to two popular automatic methods as the baselines, namely Self-Instruct (Wang et al., 2022) and Baize Self-Chat (Xu et al., 2023b). For a fair comparison, we want the generated dialogues of different methods to talk about the same things. Thus we do an additional work that we let GPT-4 generate {question, answer} pairs from the selected references and restrict the answers to the questions to be found or inferred from the references. Given a selected reference, for Self-Instruct, we follow the Alpaca (Taori et al., 2023) that we randomly select three {question, answer} pairs (from other references) as few-shot examples and add the final question from the selected reference at the end of the model input. And we let the model respond to the final question. For Baize, we use the question generated from the selected reference as the seed following the way that Baize uses questions in Quora as seeds. For RefGPT, we directly use the selected reference to generate. In practice, we select 1000 passages from Wikipedia as the references to generate 1000 seed {question, answer} pairs using the GPT-4. And we generate the dialogues using these three methods with GPT-3.5-turbo for the experiment. In Table 3, we give examples of evaluating the truthfulness of the dialogues generated by RefGPT. And more examples of Self-Instruct and Baize can be seen in Appendix C.

For human evaluation for a small sample, we randomly sample 50 English dialogues each for Alpaca, Baize, and RefGPT about factual knowledge. And 2 humans evaluate the truthfulness of the dialogues according to the references.

For automatic evaluation for a large range, in order to let GPT-4 check the factual errors without suffering from the model hallucination, we need a reference for GPT-4 to refer like RefGPT.

Therefore, as shown in Figure 2, we let GPT-4 check if the generated dialogue accords with the reference. If the generated dialogue does not align with the reference, it indicates the presence of factual errors.

### 5.1.2 Result

We use accuracy to measure the truthfulness in the evaluation process, which is the proportion of the number of dialogues without factual errors in the total of 1000 generated dialogues. In Table 2, to our surprise, we can see that Self-Instruct and Baize Self-Chat have a striking number of factual errors in the generated dialogues on both human and GPT-4 evaluations. As the dialogues generated by Baize are multi-turn, they are more likely to contain factual errors and thus have a lower truthfulness score of 47.2. By contrast, RefGPT has a truthfulness score of 97.5 with merely no factual errors. This also implicitly indicates that a model like GPT-3.5-turbo already has the ability to generate the dialogues strictly conforming to the references rather than modifying with hallucination. Another method called UltraChat (Ding et al., 2023) in Table 2 is not included, as the code has not been open-source at the time we write this paper.

## 5.2 Further Analysis

In this section, we explore the potential influence of the reference and customization on the generated dialogues by RefGPT. For each setting in the following experiments, we generate 1000 dialogues using GPT-3.5-turbo.

### 5.2.1 Dialogue Quality

As RefGPT generates the dialogue according to the reference, the reference has a significant impact on the quality of generated dialogues. We use the evaluation method mentioned in Sec 5.1 to evaluate

Table 3: Examples of truthfulness evaluation on the generated dialogues by RefGPT. We prompt GPT-4 to give the explanation first and then output the judgment.

| Reference | Dialogue | Explanation | Result |
|---|---|---|---|
| R. J. Hampton Jr ... announced that he was transferring to the nationally-ranked basketball program of Montverde Academy in Montverde, Florida, before deciding to return to Little Elm ... | **\<user\>** I heard that R. J. Hampton Jr. is one of the talented basketball players in the NBA. Can you tell me more about his career and background? **\<assistant\>** ... he transferred to Montverde Academy, where he continued to excel and earn multiple honors ... | R. J. Hampton Jr did not transfer to Montverde Academy. He had announced his intention to transfer but ultimately decided to return to Little Elm High School ... | ✘ |
| Star Wars: Rebel Assault is a rail shooter video game developed and published by LucasArts ... | **\<user\>** What is Star Wars: Rebel Assault? **\<assistant\>** Star Wars: Rebel Assault is a famous rail shooter video game, which was developed and published by LucasArts ... | Assistant's answers accurately reflect the information provided, covering the game's significance, storyline, gameplay mode ... | ✔ |

the influence of the dialogue quality (truthfulness) in the following validations.

Table 4: The truthfulness scores of 3-turn dialogues with different ratios of reference length and noise levels.

| Ref. Ratio | Truthfulness | Noise Level | Truthfulness |
|---|---|---|---|
| 100% | 96.5 | 0% | 96.5 |
| 50% | 96.2 | 10% | 96.2 |
| 25% | 97.3 | 20% | 94.8 |

**Reference Length** As length is proportional to the amount of information the reference contains, we want to find out how the reference length will influence the truthfulness of the generated dialogues. We use the dialogue template of a 3-turn dialogue, where each utterance word count of the assistant is required to be 300 words. We experiment on different lengths of reference by the proportions: 100%, 50%, and 25% of the original required length ($3 \times 300 = 900$ words).

As shown in Table 4, it is surprising to see that the truthfulness scores do not decrease much as the reference lengths are greatly reduced. We find that the GPT-3.5-turbo chooses to reduce the length of the generated utterances to obey reference despite violating the length requirement.

**Reference Quality** The reference in RefGPT can vary from plain texts to cleaned documents in the vertical domain.

In order to quantify the influence of reference quality on dialogue quality, we experiment with different qualities of references by adding additional noise. To be specific, we use the original reference as the baseline. We use HTML labels

as noise is that many references may come from the crawled data on the websites and contain many HTML labels as noise if we do not clean the data carefully. We experiment with adding 10% and 20% nonsense HTML labels as the noise.

As we can see in Table 4, the truthfulness of the generated dialogues only slightly decreases because of the additional noise. This indicates the good robustness of generating truthful dialogues even with GPT-3.5-turbo.

### 5.2.2 Dialogue Structure

During post-processing of the generated dialogues of RefGPT, we find that the input length (related to reference length) and output length (related to the required word count) will influence the success rate of obeying the dialogue template. In order to evaluate the customization ability of RefGPT, we do experiments on generating 3-turn and 5-turn dialogues. As the input length (reference length) is also determined by the required word count, we experiment with different word counts of 100, 300, and 600 for each assistant utterance to verify the success rate of obeying the dialogue template.

Table 5: The success rates (%) of obeying the dialogue templates with different word count settings for 3-turn and 5-turn dialogues.

| Word Count | Turn | w/ \</chat\> | w/o \</chat\> |
|---|---|---|---|
| 100 | 3 / 5 | 97.4 / 94.8 | 93.5 / 91.4 |
| 300 | 3 / 5 | 94.6 / 90.1 | 91.3 / 88.5 |
| 600 | 3 / 5 | 93.2 / 86.5 | 88.4 / 70.4 |

From Table 5, we can see that dialogues with fewer tokens to generate (fewer words in assistant utterances and fewer turns) will lead to better

control over the dialogue structure with a higher success rate. We further observe that if the ending mark </chat> is successfully generated, the dialogues are more likely to obey the dialogue template with the correct number of turns.

## 6 Conclusion

We present RefGPT, a new method that generates truthful and customized multi-turn dialogues using LLMs like GPT-3.5/GPT-4. Incorporating a reliable reference, RefGPT minimizes hallucination and untruthful content generation. RefGPT also allows for dialogue customization in structure, style, and content, making it flexible to generate dialogues with diversity. On the basis of RefGPT, we also use GPT-4 to construct two new multi-turn dialogue datasets, RefGPT-Fact and RefGPT-Code, based on the online encyclopedia websites and Github repositories. These datasets also showcase RefGPT's significant potential for developing dependable, domain-specific dialogue data required by specialized chatbots and other natural language processing applications.

## Limitations

RefGPT can only strictly generate the dialogues conforming to the references even though the reference itself may have factual errors. Furthermore, the generated dialogues can not be avoided to be influenced by the biases from the references. Thus the datasets RefGPT-Fact and RefGPT-Code may have factual errors and typos from Wikipedia, or bugs and malicious program codes from Github repositories.

LLMs like GPT-3.5/GPT-4 have their own biases, which will also have reflections in the dialogues generated by RefGPT.

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

# A  Dataset Card

## A.1  RefGPT-Fact

RefGPT-Fact is a dataset comprising 100k multi-turn dialogues focusing on factual knowledge. There are two versions, with the English version containing 50k dialogues based on the English Wikipedia, while the Chinese version consists of 50k dialogues sourced from the widely-used Chinese online encyclopedia, Baidu Baike.

Since most of the passages in the English Wikipedia and Baidu Baike are written by individuals or unofficial organizations, many of the passages are not commonly seen in everyday life. We use GPT-3.5-turbo API to quickly filter out the uncommon passages by asking it "Do you know xxx? If yes, return <yes>. If no, return <no>.", where xxx is the title of the passage[2].

## A.2  RefGPT-Code

RefGPT-Code is a comprehensive dataset that consists of 76k multi-turn dialogues on programming, including 37k English and 39k Chinese dialogues. As illustrated in Figure 3, it encompasses a wide range of coding scenarios about discussion, creation, and bug fixing using various programming languages. The dataset utilizes the public Github dataset available on Google BigQuery, with no overlapping data between the two languages.

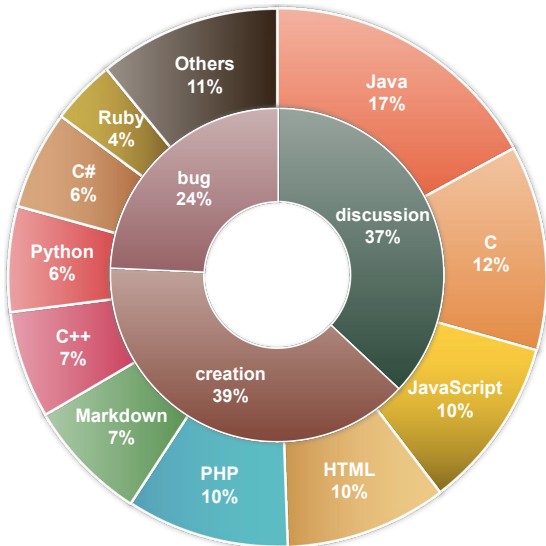

Figure 3: Composition of RefGPT-Code Dataset including English and Chinese.

---

[2]This method is based on knowledge of GPT-3.5-turbo, where recall rate is limited. A more recommended way is using the access rate to filter out the uncommon passages.

Table 6: An example of the prompt for generating the English RefGPT-Fact data. **{dialogue_template}** is replaced by the dialogue template mentioned in Sec 3.3.

---

##Provided Information## **{reference}** Based on the ##Provided Information## above and its relevant topic, expand it into a multi-round conversation. The conversation requires you to act as the chatbot Assistant and interact with a human, helping to solve the requests raised by the human. The human will ask multiple various questions/requests to the Assistant based on the information above (but the conversation should not include expressions like "according to the above information"), and the subsequent questions/requests will be a follow-up based on the previous conversation history. For every reasonable question/request posed by Human, Assistant should provide as detailed an answer as possible, offering further explanations or examples. For unreasonable requests from Human (those that are harmful to society, immoral, or illegal), Assistant will refuse to answer and explain the reason for not answering, while also providing reasonable advice to avoid such actions.

#Conversation Plan# Example: "<chat><Human 1>:(Word count requirement: x words)XXX <Assistant 1>: (Word count requirement: x words) XXX <Human 2>:(Word count requirement: x words)XXX <Assistant 2>: (Word count requirement: x words) XXX </chat>", "XXX" is the requirement for the current conversation content of that role, and "(Word count requirement: x words)" specifies the minimum word count requirement for utterance of Human or Assistant. It must be noted: the conversation starts with <chat> as the beginning of the multi-round conversation and ends with </chat> as the end of the multi-round conversation. The following conversation follows this #Conversation Plan# and word count requirements: "**{dialogue_template}**", a total of **{number_of_turns}** turns of conversation.

---

**{dialogue_template}** <chat><Human 1>:(word count: 100 words)asks a question <Assistant 1>:(word count: 200 words)answers [+detailed explanation] <Human 2>:(word count: 150 words)further asks from the perspective of real life <Assistant 2>:(word count: 100 words)answers [+detailed explanation] <Human 3>:(word count: 50 words)further asks a question <Assistant 3>:(word count: 150 words)answers [+detailed explanation] </chat>

---

Table 7: An example of the prompt for generating the English RefGPT-Code-ds data.

---

##Provided Information## **{reference}**  Based on the ##Provided Information## above and its relevant topic, expand it into a multi-round conversation. The conversation requires you to act as the chatbot Assistant and interact with a human, helping to solve the requests raised by the human. The human will ask multiple various questions/requests to the Assistant based on the information above (but the conversation should not include expressions like "according to the above information"), and the subsequent questions/requests will be a follow-up based on the previous conversation history.  For every reasonable question/request posed by Human, Assistant should provide as detailed an answer as possible, offering further explanations or examples.  For unreasonable requests from Human (those that are harmful to society, immoral, or illegal), Assistant will refuse to answer and explain the reason for not answering, while also providing reasonable advice to avoid such actions.

#Conversation Plan# Example: "<chat><Human 1>:(Word count requirement: x words)XXX <Assistant 1>: (Word count requirement: x words) XXX <Human 2>:(Word count requirement: x words)XXX <Assistant 2>: (Word count requirement: x words) XXX </chat>", "XXX" is the requirement for the current conversation content of that role, and "(Word count requirement: x words)" specifies the minimum word count requirement for utterance of Human or Assistant. It must be noted: the conversation starts with <chat> as the beginning of the multi-round conversation and ends with </chat> as the end of the multi-round conversation. The following conversation follows this #Conversation Plan# and word count requirements: "**{dialogue_template}**", a total of **{number_of_turns}** turns of conversation.

---

**{dialogue_template}**  <chat><Human 1>:(word count: 50 words)makes a request about writing the code <Assistant 1>:(word count: 250 words)answers [+detailed explanation] and give code examples <Human 2>:(word count: 100 words)asks in a young person's tone about further modifying the code <Assistant 2>:(word count: 300 words)answers [+detailed explanation] and give code examples <Human 3>:(word count: 20 words)asks from the perspective of real life about further how to use the code <Assistant 3>:(word count: 250 words)answers [+detailed explanation] and give code examples </chat>

Table 8: An example of the prompt for generating the English RefGPT-Code-cr data.

---

##Provided Information## **{reference}** Based on the ##Provided Information## above and its relevant topic, expand it into a multi-round conversation. Human has an idea / requirement / task / assignment / problem / difficulty related to the above code and wants to solve it with a computer program, but doesn't know how to do it. But Human doesn't know that the above code exists, so it can't be mentioned in conversation. Assistant needs to organize the above code into answers (which cannot be found by Human) according to Human's ideas, write specific program code for Human and explain it in detail so that Human's ideas can be realized. Based on this idea, Human would ask multiple questions and requests for specific code written by the Assistant, which will be follow-ups based on the previous conversation history. For unreasonable requests from Human (those that are harmful to society, immoral, or illegal), Assistant will refuse to answer and explain the reason for not answering, while also providing reasonable advice to avoid such actions.

#Conversation Plan# Example: "<chat><Human 1>:(Word count requirement: x words)XXX <Assistant 1>:(Word count requirement: x words) XXX <Human 2>:(Word count requirement: x words)XXX <Assistant 2>: (Word count requirement: x words) XXX </chat>", "XXX" is the requirement for the current conversation content of that role, and "(Word count requirement: x words)" specifies the minimum word count requirement for utterance of Human or Assistant. It must be noted: the conversation starts with <chat> as the beginning of the multi-round conversation and ends with </chat> as the end of the multi-round conversation. The following conversation follows this #Conversation Plan# and word count requirements: "**{dialogue_template}**", a total of **{number_of_turns}** turns of conversation.

---

**{dialogue_template}** <chat><Human 1>:(word count: 50 words)asks with curiosity about creating the code <Assistant 1>:(word count: 300 words)answers [+detailed explanation] and give code examples <Human 2>:(word count: 100 words)asks a question about further using the code <Assistant 2>:(word count: 250 words)answers [+detailed explanation] and give code examples <Human 3>:(word count: 150 words)asks a question about further explaining the code <Assistant 3>:(word count: 300 words)answers [+detailed explanation] and give code examples <Human 4>:(word count: 50 words)expresses his/her needs and asks the Assistant for help about further using the code <Assistant 4>:(word count: 200 words)answers [+detailed explanation]</chat>

Table 9: An example of the prompt for generating the English RefGPT-Code-bg data.

---

##Provided Information## **{reference}**  Based on the ##Provided Information## above and its relevant topic, expand it into a multi-round conversation.  Human will write a piece of code with bugs based on the given code above (however, Human needs to hide the presence of the given code in the conversation, and it cannot be mentioned). They will then ask Assistant for help in fixing the bugs. Assistant needs to identify the mistakes in Human's code based on the given code above (but given code cannot be discovered by Human, and it cannot be mentioned in the conversation) and provide detailed explanations on how to fix the bugs, along with more explanations or examples if necessary. Afterward, Human and Assistant will continue the conversation around this code. For unreasonable requests from Human (those that are harmful to society, immoral, or illegal), Assistant will refuse to answer and explain the reason for not answering, while also providing reasonable advice to avoid such actions.

#Conversation Plan# Example: "<chat><Human 1>:(Word count requirement: x words)XXX <Assistant 1>: (Word count requirement: x words) XXX <Human 2>:(Word count requirement: x words)XXX <Assistant 2>: (Word count requirement: x words) XXX </chat>", "XXX" is the requirement for the current conversation content of that role, and "(Word count requirement: x words)" specifies the minimum word count requirement for utterance of Human or Assistant. It must be noted: the conversation starts with <chat> as the beginning of the multi-round conversation and ends with </chat> as the end of the multi-round conversation. The following conversation follows this #Conversation Plan# and word count requirements: "**{dialogue_template}**", a total of **{number_of_turns}** turns of conversation.

---

**{dialogue_template}**  <chat><Human 1>:(word count: 500 words)asks from the perspective of real life about writing a piece of code with bugs and show the detailed code <Assistant 1>:(Word count: 250 words)answers [+detailed explanation] and tell Human about the error location in the code, then provide a correct piece of code <Human 2>:(word count: 100 words)makes a request about further using the code <Assistant 2>:(Word count: 200 words)answers [+detailed explanation] and give code examples <Human 3>:(word count: 50 words)asks with curiosity about further explaining the code <Assistant 3>:(Word count: 250 words)answers [+detailed explanation]</chat>

# B  Dataset Examples

Table 10: An example of Chinese RefGPT-Fact.

**Reference**

北上广深 北上广深是指中国大陆地区经济实力最强的四座城市为北京、上海、广州、深圳。这四座城市的综合实力在中国大陆地区处于最领先的层次。中文名 北上广深 外文名 Beijing-Shanghai-Guangzhou-Shenzhen 别 名 一线城市 城市数 4 北上广深概要 北上广深指的是北京、上海、广州、深圳。这4个城市在中国大陆地区城市中的综合实力和竞争力相对处于最领先的层次，又被称作一线城市。拥有雄厚的经济基础，以及可观的政治资源，对周边多个省份具有辐射能力，有雄厚的教育资源、深厚的文化，还具有繁华、美丽的夜景，极为便利的交通和独特的城市魅力。 北上广深城市概况 北上广深北京 北京（Beijing），简称京，中华人民共和国首都、直辖市、国际大都市、国家中心城市、超大城市，全国政治中心、文化中心、国际交往中心、科技创新中心，经济与金融的管理中心和决策中心，是中国共产党中央委员会、中华人民共和国中央人民政府和中华人民共和国全国人民代表大会的办公所在地。北京历史悠久，文化灿烂，是首批国家历史文化名城、中国八大古都之一和世界上拥有世界文化遗产数最多的城市，3060年的建城史孕育了故宫、天坛、八达岭长城、颐和园等众多名胜古迹。早在七十万年前，北京周口店镇地区就出现了原始人群部落"北京人"。公元前1045年，北京成为蓟、燕等诸侯国的都城。公元938年以来，北京先后成为辽陪都、金上都、元大都、明清国都、民初京兆。1949年10月1日成为中华人民共和国首都。2015年末，北京全市常住人口2170.5万人，比2014年末增加18.9万人。其中，常住外来人口822.6万人，占常住人口的比重为37.9%。2015年北京市实现地区生产总值22968.6亿元，比2014年增长6.9%。2015年7月31日，国际奥林匹克委员会主席托马斯·巴赫宣布北京携手张家口获得2022年冬季奥林匹克运动会的举办权。北京由此成为全球首个既举办过夏季奥林匹克运动会又即将举办冬季奥林匹克运动会的城市。 北上广深上海 上海（Shanghai），简称"沪"或"申"，中华人民共和国直辖市，超大城市，国际大都市、中国国家中心城市，中国的经济中心、贸易中心、航运中心.首批沿海开放城市。地处长江口，隔东中国海与日本九州岛相望，南濒杭州湾，西与江苏、浙江两省相接。

---

**<user>** 北上广深是指哪四个城市？
**<assistant>** 北上广深是指中国大陆地区经济实力最强的四座城市，它们分别是北京、上海、广州和深圳。这四座城市的综合实力在中国大陆地区处于最领先的层次，且被称作一线城市。由于这些城市拥有雄厚的经济基础、政治资源以及可观的繁荣程度，它们对周边省份具有巨大的辐射能力。

**<user>** 北京在中国有哪些重要的地位？
**<assistant>** 北京，作为中华人民共和国首都，拥有在中国非常重要的地位。首先，它是全国政治中心，是中国共产党中央委员会、中华人民共和国中央人民政府和全国人民代表大会的办公所在地。其次，它也是中国的文化中心，是首批国家历史文化名城、中国八大古都之一，具有深厚的文化底蕴。此外，北京还是国际交往中心、科技创新中心以及经济与金融的管理中心和决策中心。

**<user>** 你能否详细讲一下2015年北京市的经济状况？
**<assistant>** 根据2015年的数据，北京市实现地区生产总值(GDP)为22968.6亿元，比2014年增长了6.9%。同年末，北京全市常住人口为2170.5万人，比2014年末增加了18.9万人。其中，常住外来人口为822.6万人，占常住人口的比重为37.9%。这些数据反映了北京市2015年的经济状况以及人口变化情况。

**<user>** 上海市有什么特点？
**<assistant>** 上海，简称"沪"或"申"，是一座具有许多特点的大都市。首先，它是中华人民共和国的直辖市，同时也是国家中心城市。其次，上海市是中国的经济中心、贸易中心和航运中心，拥有很强的经济实力。此外，上海市是首批沿海开放城市之一，地处长江口，也是一重要的地理枢纽。因为这些特点，上海在国内和国际上都具有极高的地位和影响力。

Table 11: An example of English RefGPT-Fact.

**Reference**

Joomla (), also spelled Joomla! (with an exclamation mark) and sometimes abbreviated as J!, is a free and open-source content management system (CMS) for publishing web content on websites. Web content applications include discussion forums, photo galleries, e-Commerce and user communities and numerous other web-based applications. Joomla is developed by a community of volunteers supported with the legal, organisational and financial resources of Open Source Matters, Inc. Joomla is written in PHP, uses object-oriented programming techniques and software design patterns, and stores data in a MySQL database. It has a software dependency on the Symfony PHP framework. Joomla includes features such as page caching, RSS feeds, blogs, search, and support for language internationalisation. It is built on a model–view–controller web application framework that can be used independently of the CMS. Around 6,000 extensions are available from the Joomla website, and more are available from other sources. As of 2021, it was estimated to be the fifth most used CMS on the Internet, after WordPress, Shopify, Wix and Squarespace. Overview Joomla has a web template system using a template processor. Its architecture is a front controller, routing all requests for non-static URIs via PHP which parses the URI and identifies the target page. This allows support for more human-readable permalinks. The controller manages both the frontend, public-facing view, and a backend (GUI-driven) administration interface. The administration interface (a) stores management and content information within a database, and (b) maintains a configuration file (, usually located in the file system root of the Joomla installation). The configuration file provides the connection between the server, database and file system and facilitates migrating the website from one server to another. The backend interface allows website operators to manage users, menus, extensions and web content ... (About 900 words)

<user> I recently came across joomla and i was wondering if you could tell me more about this content management system (cms). how does it compare to other cms like wordpress, shopify, wix, and squarespace? what type of users does it cater to, and what kind of applications can be built using joomla?
<assistant> Joomla is a free and open-source content management system (cms) that is widely used for creating web content on websites. it is designed to cater to users who have basic website creation skills and is ideal for building various web-based applications such as discussion forums, photo galleries, e-commerce sites, user communities and much more. joomla is developed and maintained by a community of volunteers, with open source matters, inc. offering legal, organizational, and financial resources to support the project.

Joomla is built using php, object-oriented programming techniques, and software design patterns. it stores data in a mysql database and relies on the symfony php framework. some of its features include page caching, rss feeds, blogs, search capabilities, and support for language internationalization. it also comes with a model-view-controller web application framework that can be used independently of the cms.

As of 2021, joomla is estimated to be the fifth most used cms on the internet, coming after wordpress, shopify, wix, and squarespace. with around 6,000 extensions available on the joomla website and more from other sources, it provides flexibility and extensibility to users.

Comparing it to other cms, joomla's main advantage is its open-source nature and extensive customization options through its wide range of extensions. while wordpress is more user-friendly and has a larger community, joomla strikes a balance between ease of use and flexibility. shopify, wix, and squarespace, on the other hand, are more geared towards non-technical users and have limitations in extensibility when compared to joomla.

In summary, joomla is a flexible and customizable option for users who have some website creation skills and require a cms to build diverse web applications.

<user> Thanks for explaining that. I've heard that Joomla has a unique web template system and a front-controller architecture. Can you explain how these features work together and how they benefit the user? Also, what kind of risk management, backup, and recovery options are available with Joomla?
<assistant> Joomla's web template system uses a template processor, and its architecture is based on a front-controller model. this means that all requests for non-static uris are routed through php, which parses the uri and identifies the target page. this approach allows for more human-readable permalinks, improving the user experience and search engine friendliness of a website.

Joomla's front-controller architecture manages both the frontend, public-facing view, and a backend, gui-driven administration interface. the backend allows website operators to manage users, menus, extensions, and web content. the system stores management and content information within a database and maintains a configuration file to manage the connection between the server, database, and file system.

As for risk management, backup, and recovery, joomla itself does not have built-in features for these. however, third-party extensions and standalone products are available to provide backup and recovery functionality. it is the website operator's responsibility to implement and manage these tools to ensure the integrity and security of their website.

To summarize, joomla's web template system and front-controller architecture offer improved user experience, search engine optimization, and ease of administration. although the core cms doesn't provide backup and recovery features, third-party options fill this gap and deliver comprehensive risk management and data protection capabilities.

(Two more turns are not shown because of the limited space. The whole dialogue contains about 1050 words.)

Table 12: An example of Chinese RefGPT-Code-ds.

---

**Reference**

---

```
describe Remover::Team do
  let(:github_client) { double('Github Client') }
  let(:github_team) { double('Github Team', id: 1, name: 'Owners') }
  let(:team) { Remover::Team.new(github_client, github_team) }

  describe '#name' do
    it 'returns Github team name' do
      expect(team.name).to eq(github_team.name)
    end
  end
      ...
```

---

**<user>** 在日常工作中，我发现有些Github团队可能没有实际的项目和成员，这使得管理起来非常麻烦。请问如果我想识别并删除这些无用的团队，应该如何修改这段代码？

```
describe Remover::Team do
  let(:github_client) { double('Github Client') }
  let(:github_team) { double('Github Team', id: 1, name: 'Owners') }
  let(:team) { Remover::Team.new(github_client, github_team) }

  describe '#name' do
    it 'returns Github team name' do
      expect(team.name).to eq(github_team.name)
    end
  end
      ...
```

**<assistant>** 我明白您的需求，您可以使用这段代码中的'Remover::Team'类来实现您的目标。在这个类里，提供了'#used'方法来判断该团队是否有实际的项目和成员。如果团队中没有成员和项目，该方法会返回'false'。
首先，您需要创建一个'Remover::Team'对象，传入Github客户端和Github团队实例。例如：

```
github_client = double('Github Client')
github_team = double('Github Team', id: 1, name: 'Owners')
team = Remover::Team.new(github_client, github_team)
```

之后，您可以通过'team.used'方法判断团队是否有实际的项目和成员。如果该方法返回'false'，您可以调用'team.delete_team'方法删除这个团队。例如：

```
if not team.used
    team.delete_team
end
```

**<user>** 谢谢你的解答，我明白了。那我还想请教一下，有关代码中的其他方法，如'#members_amount'和'#repositories_amount'，它们都是如何实现的？它们会返回什么结果？如果我想得到具体的成员和仓库信息，而不只是成员数量和仓库数量，我应该如何操作？
**<assistant>** 'Remover::Team'类中的'#members_amount'和'#repositories_amount'方法分别用于得到团队中成员数量和项目仓库数量。这两个方法的实现都非常简单，它们分别调用Github客户端的'team_members'和'team_repositories'方法，传入团队的ID，并获取结果集的长度。...

(Two more turns are not shown because of the limited space. The whole dialogue contains about 1400 words.)

Table 13: An example of English RefGPT-Code-ds.

**Reference**

```
# -*- coding: utf-8 -*-
import datetime
from south.db import db
from south.v2 import SchemaMigration
from django.db import models

class Migration(SchemaMigration):

    def forwards(self, orm):
        # Adding field 'Question.order'
        db.add_column(u'survey_question', 'order',
                      self.gf('django.db.models.fields.IntegerField')(default=0
                                                                     ),
                      keep_default=False)

        ...
```

**<user>**

```
# -*- coding: utf-8 -*-
import datetime
from south.db import db
from south.v2 import SchemaMigration
from django.db import models

class Migration(SchemaMigration):
        ...
```

In the context of the provided code, I see there's a Django migration for adding a color field to an Idea model. Can you explain how a migration works in Django and how to apply this specific migration to add the 'color' field to the 'Idea' model in an existing Django project?

**<assistant>** Sure! In Django, migrations are a way of managing and propagating changes made to your models (which affect the database schema) over time. A migration is essentially a set of instructions that the Django migration system uses to modify your database schema to match the current state of your model.

Applying migrations to your Django project is a two-step process:
1. Creating the migration (migration file generation): When you make changes to your model, you need to inform Django about the changes. You do this by running the 'makemigrations' command, which creates a new migration file based on the alterations in the model.
2. Applying the migration (migration execution): After generating the migration file, you need to apply it to your database to update the schema. You do this by running the 'migrate' command, which processes the migration files sequentially and updates the database schema accordingly.

In the provided code, a migration is created to add a 'color' field to the 'Idea' model. To apply this migration in your Django project, follow these steps:
1. First, ensure that the migration is in the correct location within your Django project. Migration files are typically placed within a folder named 'migrations' inside an application folder. If the Django app containing your 'Idea' model is named 'brainstorming', the migration file should be located at 'brainstorming/migrations/'.
2. Next, check if the migration has been generated by running 'python manage.py makemigrations'. This will generate new migration files based on the changes in your models. Since the code you provided already includes a migration file, this step may not be necessary. However, it's recommended to run this command to ensure all migrations are accounted for.
3. Now, apply the migration to update your database schema by running 'python manage.py migrate'. This command will process all pending migration files and apply them to the database.

By following these steps, you'll successfully apply the migration to add the 'color' field to the 'Idea' model in your existing Django project. Note that this migration is reversible, meaning you can undo the change by executing the 'python manage.py migrate <app_name> <migration_name_before_your_migration>' command. In this case, the '<app_name>' would be 'brainstorming', and '<migration_name_before_your_migration>' would be the name of the migration immediately preceding the one adding the 'color' field.

(Two more turns are not shown because of the limited space. The whole dialogue contains about 1250 words.)

Table 14: An example of Chinese RefGPT-Code-cr.

---

**Reference**

---

```
...
#include "ScriptPCH.h"
#include "zulaman.h"

...

#define SPELL_REMOVE_AMANI_CURSE 43732
#define SPELL_PUSH_MOJO 43923
#define ENTRY_FOREST_FROG 24396

class npc_forest_frog : public CreatureScript
        ...
```

---

**<user>** 我在设计一个游戏关卡，想要让玩家帮助一名被诅咒变成青蛙的角色恢复原状。游戏中，玩家可以使用去除诅咒的法术，对青蛙施法后，有一定概率会将青蛙恢复原形。如果诅咒被解除，青蛙会变回不同的角色，并且有几率掉落道具。请问如何编写这部分程序代码呢

**<assistant>** 根据您的需求，我们可以使用C++编写一个类来实现这个功能。首先，我们需要创建一个名为"npc_forest_frog"的类，该类将继承自一个外部库提供的"CreatureScript"类。我们需要在这个新类中实现青蛙的诅咒解除逻辑。在'npc_forest_frog'类中，我们需要创建一个嵌套的结构体'npc_forest_frogAI'，该结构体继承自'ScriptedAI'类。'ScriptedAI'类是游戏引擎中AI相关的基类。在这个结构体内，我们首先定义几个函数来实现AI的基本逻辑，例如"Reset"、"EnterCombat"等。

接下来，我们需要实现诅咒被解除时的行为。当法术命中青蛙时，我们需要在'SpellHit'函数中编写具体的逻辑。例如，当法术的ID为'SPELL_REMOVE_AMANI_CURSE'（您可以自定义该常量的值）时，我们检查青蛙是否处于诅咒状态。如果是，我们可以生成一个随机数，根据该随机数决定青蛙恢复成哪个角色，以及是否掉落道具。具体代码如下：

```
#include "ScriptPCH.h"
#include "zulaman.h"

#define SPELL_REMOVE_AMANI_CURSE 43732
#define SPELL_PUSH_MOJO 43923
#define ENTRY_FOREST_FROG 24396

class npc_forest_frog : public CreatureScript
{
  public:
    npc_forest_frog() : CreatureScript("npc_forest_frog") {}

    struct npc_forest_frogAI : public ScriptedAI
    {

...
```

当您编写完代码后，可以将该代码添加到游戏引擎中，并进行相应的配置以创建您想要的游戏关卡。

**<user>** 那么如果我想让掉落的道具是低概率事件并让玩家感觉更有挑战性，我应该如何调整代码呢？

**<assistant>** 要调整掉落道具的概率，您只需修改'SpellHit'函数中'rand() % 100'后面的数字。这个数字表示掉落道具的概率。例如，如果您想让掉落道具的概率降低到10%，可以将其修改为：

```
if (rand() % 100 < 10) // 10%
    DoCast(caster, SPELL_PUSH_MOJO, true);
else
    DoSpawnRandom();
```

...

(One more turns are not shown because of the limited space. The whole dialogue contains about 1150 words.)

Table 15: An example of English RefGPT-Code-cr.

**Reference**

```
package ch.zhaw.mapreduce.plugins.socket.impl;

import javax.inject.Inject;
import ch.zhaw.mapreduce.CombinerInstruction;
import ch.zhaw.mapreduce.MapInstruction;
import ch.zhaw.mapreduce.ReduceInstruction;
        ...
```

**<user>** I am working on a project that involves different tasks like mapping, reducing, and combining. I need to create task runners that can execute these tasks dynamically based on the received instructions. I also need to differentiate between mapping and reducing tasks to execute them correctly. Can you help me create a Java code that can achieve this?
**<assistant>** Certainly! Let's start by creating an interface called 'TaskRunnerFactory' with a single method 'createTaskRunner', that takes an 'AgentTask' object as input and returns a 'TaskRunner' object. Later, we'll implement the logic for differentiating between the mapping and reducing tasks. Here's the interface:

```
import javax.inject.Inject;

public final class TaskRunnerFactoryImpl implements TaskRunnerFactory {
    private   final ByteArrayClassLoader classLoader = new ByteArrayClassLoader
                                        ();
    private   final MapTaskRunnerFactory mtrFactory;
    private   final ReduceTaskRunnerFactory rtrFactory;

    @Inject
    public   TaskRunnerFactoryImpl(MapTaskRunnerFactory mtrFactory,
                                        ReduceTaskRunnerFactory rtrFactory)
                                        {
        this .mtrFactory = mtrFactory;
        this .rtrFactory = rtrFactory;
    }

    @Override
    public   TaskRunner createTaskRunner(AgentTask task) throws
                                        InvalidAgentTaskException {
        if  (task instanceof MapAgentTask) {
            // Handle MapAgentTask
        } else if (task instanceof ReduceAgentTask) {
            // Handle ReduceAgentTask
        } else {
            throw  new InvalidAgentTaskException("Cannot Handle: " + task)
                                        ;
        }
    }
}
```

In this implementation, we first check if the 'AgentTask' is an instance of 'MapAgentTask' or 'ReduceAgentTask'. Based on that, we'll execute the respective logic. We also use a custom 'ByteArrayClassLoader' to load the classes dynamically and call different getters based on the task type.

Inside the 'createTaskRunner()' method, we can now implement the logic to handle 'MapAgentTask' and 'ReduceAgentTask' instances:

```
// For MapAgentTask
MapAgentTask mt = (MapAgentTask) task;
MapInstruction mapInstr = loadClass(mt.getMapInstructionName(), mt.
                                        getMapInstruction(), MapInstruction.
                                        class);
...
```

...
(Two more turns are not shown because of the limited space. The whole dialogue contains about 800 words.)

Table 16: An example of Chinese RefGPT-Code-bg.

**Reference**

```
...
class BTree:
  def __init__(self, value):
    self .root = Node(value)
...
```

<user>
嗨Assistant，我最近在学习Python，现在遇到了一个问题。我已经写了一段代码，但是它运行不起来，能帮忙看一下是哪里出了问题吗？代码如下：

```
...
class BTree:
...
  def insert(self, value):
    def helper(cur_node, value):
      if value > cur_node.value:
        if cur_node.right:
          cur_node = cur_node.right
          helper (cur_node, value)
        else :
          cur_node .right = Node(value)
      else :
        if cur_node.left:
          cur_node = cur_node.left
          helper (cur_node, value)
        else :
          cur_node .left = Node(value)
    helper (self.root, value)
...
```

这段代码是我创建了一个二叉搜索树，并插入了一些节点，但是当我运行这段代码时，它卡住了。我想知道错误在哪里，以及如何修复它。谢谢！

<assistant> 很高兴能帮助你解决问题。首先，我注意到你的代码中存在一个关于递归调用的问题。在'insert'方法的'helper'函数中，你需要在递归调用之前使用return语句。请注意，在调用 'helper()'函数时，应使用'return helper(cur_node, value)'而不是'helper(cur_node, value)'，这样递归调用时才能正确返回。试试以下代码：

```
...
class BTree:
...
  def insert(self, value):
    def helper(cur_node, value):
      if value > cur_node.value:
        if cur_node.right:
          cur_node = cur_node.right
          return helper(cur_node, value)
        else :
          cur_node .right = Node(value)
      else :
        if cur_node.left:
          cur_node = cur_node.left
          return helper(cur_node, value)
        else :
          cur_node .left = Node(value)
    return helper(self.root, value)
...
```

现在这段代码应该可以正常工作了。如果你还有其他问题，请随时告诉我。

(Three more turns are not shown because of the limited space. The whole dialogue contains about 1300 words.)

Table 17: An example of English RefGPT-Code-bg.

---

**Reference**

---

```
...
(function($) {
  var methods = {
    init : function(options) {
      var defaults = {

        element : {
          selector  : "",
          shadowSize  : "30",
          shadowColor : "#222D3A",
          shadowFade  : "0.3",
          shadowAlpha : "0.03",
        },
...
```

---

**\<user\>** Hey, I was trying to implement a long shadow effect on a text element using jQuery, but I cannot seem to get the desired effect. I've written this code, but it seems to have some bugs. Can you help me fix it?
Here's the code:

```
...
function applyLongShadow(o) {
...
    for (var i = 0; i <= o.textElement.shadowSize; i++) {
        var x = o.textElement.shadowSize - i;
        var y = o.textElement.shadowSize - i;
        var shadow = x + "px " + y + "px ";

        if (o.textElement.shadowFade && i < fade) {
            shadow += xcolor.gradientlevel(darkerTX, txc, fadeText-i, fadeText)
                                                      ;
...
```

**\<assistant\>** I see that you would like to apply a long shadow effect on a text element using jQuery. I noticed a couple of issues in your code that might be causing the undesired effect.

Firstly, you are using 'xcolor.gradientlevel()' in your code, but you haven't included the 'xcolor' library or defined the variables 'darkerTX', 'txc', and 'fadeText'. You can instead use the '$.$xcolor.gradientlevel()$'$method from the$'$.xcolor' library, and make sure to include the library in your project.

Here's a corrected version of your code:

```
...
function applyLongShadow(o) {
...
    for (var i = 0; i <= o.textElement.shadowSize; i++) {
        var x = o.textElement.shadowSize - i;
        var y = o.textElement.shadowSize - i;
        var shadow = x + "px " + y + "px ";

        if (o.textElement.shadowFade && i < fade) {
            var tweenedTX = $.xcolor.gradientlevel(darkerTX, txc, fade - i, fade
                                                      );
...
```

Please make sure to include the '$.xcolor' library in your project, and let me know if you need further assistance.

(One more turns are not shown because of the limited space. The whole dialogue contains about 550 words.)

# C   Truthfulness Evaluation Examples

Table 18: An Example of truthfulness evaluation on the generated dialogues by Self-Instruct (Wang et al., 2022).

| Reference | Dialogue | Explanation | Result |
|---|---|---|---|
| The fifth series of British talent competition programme Britain's Got Talent was broadcast on ITV ...   As Piers Morgan had departed from the programme the previous year, and the schedule of Simon Cowell made him unable to attend auditions, the producers arranged for Amanda Holden to be joined by David Hasselhoff and Michael McIntyre on the judging panel. Due to Hasselhoff's schedule during filming of the auditions, Louis Walsh stepped in as a guest judge for the sessions he could not attend ... | **<user>** Who were the judges for the 5th series of Britain's Got Talent, and what changes occurred in the judging panel that year? **<assistant>** The judges for the 5th series of Britain's Got Talent were Simon Cowell, Amanda Holden, Piers Morgan, and Kelly Brook, with Ant and Dec as hosts. That year, Kelly Brook was replaced by a rotating panel of guest judges due to her absence after just six days of filming. | <1> The Assistant incorrectly states that the judges for the 5th series of Britain's Got Talent were Simon Cowell, Amanda Holden, Piers Morgan, and Kelly Brook. The correct judges were Amanda Holden, David Hasselhoff, and Michael McIntyre, with Simon Cowell only present for the live episodes. Piers Morgan had departed from the program the previous year, and Kelly Brook was not involved at all. <2> The answer claims that Kelly Brook was replaced by a rotating panel of guest judges, but this is incorrect. In the 5th series, there were no guest judges other than Louis Walsh, who temporarily replaced David Hasselhoff when he could not attend the London auditions. | ✘ |
| A consumer economy describes an economy driven by consumer spending as a percent of its gross domestic product, as opposed to the other major components of GDP (gross private domestic investment, government spending, and imports netted against exports). In the U.S., it is usually said by economists, including in Henry Hazlitt's Ëconomics in One Lesson"that 70% of spending is consumer-based, but this number is disputed by economists like Businessweek columnist Michael Mandel ... | **<user>** What are the main components of gdp and how does a consumer economy fit into this? **<assistant>** The main components of GDP are consumption, investment, government spending, and net exports. A consumer economy is an economy that is driven by the spending and saving behavior of individuals. Consumption is the largest component of GDP in most countries, including the United States. Therefore, the behavior of consumers is a major driver of economic growth. | The Assistant's answer correctly identifies the main components of GDP, which are consumption, investment, government spending, and net exports. The answer also accurately explains how a consumer economy fits into this, stating that consumption is the largest component of GDP in most countries and thus the behavior of consumers drives economic growth. | ✔ |

Table 19: An Example of truthfulness evaluation on the generated dialogues by Baize (Xu et al., 2023b).

| Reference | Dialogue | Explanation | Result |
|---|---|---|---|
| Scoat Fell is a fell in the western part of the English Lake District. It stands at the head of the Mosedale Horseshoe with its back to Ennerdale. Paths lead to Scoat Fell from Ennerdale over Steeple, from Wasdale over Red Pike, and along the ridge from Pillar. Topography The Western Fells occupy a triangular sector of the Lake District, bordered by the River Cocker to the north east and Wasdale to the south east. Westwards the hills diminish toward the coastal plain of Cumberland. At the central hub of the high country are Great Gable and its satellites, while two principal ridges fan out on either flank of Ennerdale, the western fells in effect being a great horseshoe around this long wild valley. Scoat Fell is on the southern arm ... | **<user>** Can you tell me the location of Scoat Fell in the English Lake District? **<assistant>** Sure! Scoat Fell is located in the western part of the English Lake District, specifically in the Wasdale area. It is one of the highest peaks in the Lake District, with a height of 841 metres or 2,759 feet.

**<user>** How can I access Scoat Fell? **<assistant>** There are several ways to access Scoat Fell. One of the most popular routes is to start from the Wasdale Head village and hike up via the Brown Tongue path. Another option is to start from the Lake District Youth Hostel in Eskdale and hike up via the path that leads to Burnmoor Tarn.

**<user>** Are there any other peaks near Scoat Fell that I can visit? **<assistant>** Yes, there are several peaks near Scoat Fell that are worth visiting. One of them is Scafell Pike, which is the highest mountain in England and is located only a few miles to the east of Scoat Fell. Other peaks nearby include Pillar, Haycock, and Red Pike. | <1> The location of Scoat Fell is mentioned inaccurately as being in the Wasdale area, but it is actually in the western part of the English Lake District at the head of the Mosedale Horseshoe with its back to Ennerdale. <2> The provided access routes to Scoat Fell are incorrect. According to the provided information, the paths to Scoat Fell are from Ennerdale over Steeple, from Wasdale over Red Pike, and along the ridge from Pillar. | ✘ |
| Morchella esculenta, (commonly known as common morel, morel, yellow morel, true morel, morel mushroom, and sponge morel) is a species of fungus in the family Morchellaceae of the Ascomycota. It is one of the most readily recognized of all the edible mushrooms and highly sought after. Each fruit body begins as a tightly compressed, grayish sponge with lighter ridges, and expands to form a large yellowish sponge with large pits and ridges raised on a large white stem. The pitted yellow-brown caps measure broad by tall, and are fused to the stem at its lower margin, forming a continuous hollow. The pits are rounded and irregularly arranged. The hollow stem is typically long by thick, and white to yellow. The fungus fruits under hardwoods and conifers during a short period in the spring, depending on the weather, and is also associated with old orchards, woods and disturbed grounds ... | **<user>** Can you tell me about morchella esculenta and its main features? **<assistant>** Morchella esculenta is a species of edible mushroom, also known as the morel mushroom. Its main features include a distinctive honeycomb appearance and a meaty texture.

**<user>** Where can morchella esculenta be found? **<assistant>** Morchella esculenta can typically be found in areas with rich soil, such as forests and grasslands. It is commonly found in North America and Europe during the spring season.

**<user>** Are there any other types of morel mushrooms besides morchella esculenta? **<assistant>** Yes, there are several other types of morel mushrooms, including Morchella conica and Morchella deliciosa. However, Morchella esculenta is the most commonly consumed morel mushroom.

**<user>** Can you describe the flavor of morchella esculenta? **<assistant>** Morchella esculenta has a complex umami flavor that is often described as nutty, earthy, and slightly smoky. It pairs well with butter and herbs, making it a popular ingredient in many recipes. | The dialogue correctly describes the main features of Morchella esculenta, its habitat, alternative types of morel mushrooms, and the flavor profile of Morchella esculenta. The answers provided by the AI are consistent with the provided information, and there are no factual errors. | 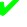 |