# OpenReview forum: "RefGPT: Dialogue Generation of GPT, by GPT, and for GPT"
_EMNLP/2023/Conference — EMNLP 2023 Findings_

### Official Review · Reviewer_7XKU · 2023-08-02

**Typos Grammar Style And Presentation Improvements:** 1. Line3 fine-tuning with
2. Line 40 …
**Soundness:** 3

**Excitement:**

3: Ambivalent: It has merits (e.g., it reports state-of-the-art results, the idea is nice), but there are key weaknesses (e.g., it describes incremental work), and it can significantly benefit from another round of revision. However, I won't object to accepting it if my co-reviewers champion it.

**Paper Topic And Main Contributions:**

This paper focuses on the instruction data construction problem. Similar to the previous method, like Self-Instruct, it utilized powerful LLMs (GPT3.5 and GPT-4) to generate dialogue data in a bootstrapping manner. The main difference is that it adds some reference messages as an additional information source for the input prompt to ensure the LLM can generate more authentic and truthful dialogue data.  Experiments show that the proposed RefGPT method can gather high truthfulness data and also more customized data via detailed prompt design than baselines. In the end, two relatively large instruction datasets RefGPT-Fact and RefGPT-Code are proposed for the LLM instruction tuning.

**Questions For The Authors:**

1. There are some overclaims in the paper. line 13 "without worrying about factual errors caused by the model hallucination"  line70 "LLMs will be prompted not to rely on their own knowledge to generate the dialogues, thus resolving the hallucination issue.". Even given the reference as input, it still can not guarantee LLMs to generate perfect answers. Just as the paper mentioned in 5.2.2, even the </chat> marker can not be guaranteed to be generated.

2. In Table 4. why 25% Ref. Ratio has the largest truthfulness? Also, why the 100% ref.Ratio does not have the same number with 0% Noise Level. It seems that the first row should be 100% Ref.Ratio and 0% Noise Level.


**Reasons To Accept:**

1. Contributing two dialogue instruction tuning datasets (total 176k sessions) to the NLP community for potentially better LLM fine-tuning. The dataset is generated by GPT-4, using the online encyclopedia websites and Github repositories as the input references to generate more truthful dialogues.

2. The prompt design experience about more fine-grained control in dialogue structure, number of turns, utterance length, and dialogue styles may bring some insights into the problem of locally customized LLM generation.

3. Proposing an automatic evaluation framework of using GPT-4 as the referee to evaluate instruction data.  Given the ground-truth answer source (i.e., snippets of reference), GPT-4 can determine which dialogue is better and scale up to much larger benchmarks.


**Reasons To Reject:**

1. The biggest problem is that whether the proposed dialogue instruction data can really boost the performance of the fine-tuned LLMs is unclear.  The comparison experiment of tuning open-source LLMs like LLaMa on self-instruct and RefGPT is lacking. Better quality instruction data does not guarantee the tuned LLMs can avoid hallucination since the LLM still relies on itself to generate during the inference. A stronger baseline would be an LLM that is fine-tuned on self-instruct during training and performs BM25-retrieved conditioned generation from the same online source as RefGPT during testing. If the author can verify that an LLM that is fine-tuned on RefGPT datasets is on-par with or even better than that strong baseline, this paper would be much more convincing and complete.

2. Lacking some human evaluation for the RefGPT. Although it is intuitive that RefGPT is more faithful and reliable than self-instruct and Baize, but the quality of the dataset is only proved good by GPT-4, another model. There could be some bias in using a model to judge; for example, it tends to select an answer that has a short repetition of the source reference, which may still not be the true answer to the question or hurt the answer variety. There still needs more experiments, esp. some human studies, to show the quality of RefGPT.

3. Adding more external textual information as conditional input to generate less hallucinated data for LLMs is a common practice, so the methodology for this paper is not novel. The main contribution of the paper is actually the dataset, but the authors need more experiments to show its efficacy.

**Reproducibility:**

5: Could easily reproduce the results.

**Reviewer Confidence:**

4: Quite sure. I tried to check the important points carefully. It's unlikely, though conceivable, that I missed something that should affect my ratings.

---

> ### Author Rebuttal · Authors · 2023-08-28
>
> Thanks so much for your insightful comments!
>
> W1: Whether the proposed datasets can boost the performance is unclear. The comparison of tuning open-source LLMs is lacking. Better quality instruction data does not guarantee that tuned LLMs can avoid hallucination. A stronger baseline, that a LLM is fine-tuned on self-instruct during training and performs BM25-retrieved conditioned generation from the same online source as RefGPT during testing, is needed.
>
> A1: We agree that less hallucinated data can not guarantee the tuned LLM can avoid hallucination. However, the high-quality RefGPT data alleviates the harm of tuning with hallucinated data to some extent.
>
> **We supplement the strong baseline as you mention.** We fine-tune the llama-13b with RefGPT-Fact and with Alpaca separately. The strong baseline is the llama-13b model tuned with Alpaca and using the BM25 to find the reference as context. 2 humans evaluate 50 English questions in the RefGPT-Fact dataset on these two models. These two models both use a temperature of 0.1, top_p of 0.75, and top_k of 40 for generation. The Alpaca model with BM25 retrieval has an acc of 76.0 and RefGPT has an acc of 82.0. We observe that the retrieval-based method sometimes retrieves the wrong references so the model has to recite its own knowledge. And RefGPT model uses its memory in the fine-tuning state to answer the question but sometimes generates with hallucination. The results show that RefGPT can alleviate the hallucination on factual knowledge and is slightly better than the naive retrieval-based method.
>
>
>
> W2: Lacking some human evaluation for the RefGPT.
>
> A2: Thanks for pointing this out. In the evaluation, we use the GPT-4 to compare if the assistant's responses are contradictory to the references, instead of letting GPT-4 judge the truthfulness of the responses with its own knowledge base.
>
> **We have supplemented the human evaluation following a similar pattern.** We randomly sample 50 English dialogues each for Alpaca, Baize, and RefGPT about factual knowledge. And 2 humans evaluate the truthfulness of the dialogues according to the references. The accuracies of Alpaca, Baize, and RefGPT are 54.0, 50.0, and 98.0. The results by humans have a similar trend as the GPT-4 evaluation (50.2, 47.2, 97.5).
>
>
>
> W3: Using the external text to generate less hallucinated data is a common practice. The main contribution is the dataset, but the authors need more experiments to show its efficacy.
>
> A3: RefGPT is not only for generating less hallucinated data but also has the potential to generate dialogues of any vertical domain, e.g. coding scenario or a shopping assistant, by utilizing domain-specific documents as references. Besides utilizing the external text, RefGPT enables customization to generate different types of dialogues even with the same reference. For a piece of code, RefGPT can derive different scenarios of code creation, debugging, discussion, and so on.
>
> As collecting code-related dialogues needs a large amount of domain knowledge, the RefGPT-Code dataset from real-world code data has intuitively better quality than simply collecting the synthetic code data from GPT-4. We will also train a model for the code scenario using the RefGPT-Code later to show more efficacy.
>
>
>
> Q1: Overclaims of resolving the model hallucination in dialogue generation.
>
> A4: We agree that GPT-4 can not always follow the references to generate the dialogues correctly thus our claims will be revised according to your advice.
>
>
>
> Q2: In Table 4. why 25% Ref. Ratio has the largest truthfulness? Why the 100% Ref.Ratio does not have the same number with a 0% Noise Level. It seems that the first row should be 100% Ref.Ratio and 0% Noise Level.
>
> A5: A smaller Ref ratio (25%) has a shorter context and GPT tends to generate a shorter dialogue, which makes it easier for GPT to follow the reference, thus having the largest truthfulness.
>
> The noise level experiments are done separately on different settings (150 words each turn) in the paper from the Ref ratio experimental settings (300 words), thus there is a difference between 100% Ref.Ratio and 0% Noise Level. We supplement the experiments with the same settings (300 words) and the results are shown below:
>
> | Noise Level | Truthfulness |
> | ----------- | ------------ |
> | 0           | 96.5         |
> | 10          | 96.2         |
> | 20          | 94.8         |
> | 30          | 93.3         |
> | 50          | 90.1         |
> | 70          | 86.2         |
>
>
>
> Q3: Typos Grammar Style And Presentation Improvements
>
> A6: Thanks a lot for pointing these out. We will revise our paper according to your advice to make the paper express more clearly.

---

### Official Review · Reviewer_8x7N · 2023-08-04

**Soundness:** 4

**Excitement:**

4: Strong: This paper deepens the understanding of some phenomenon or lowers the barriers to an existing research direction.

**Paper Topic And Main Contributions:**

This manuscript proposes RefGPT to generate enormous truthful and customized dialogues without worrying about factual errors caused by the model hallucination.

**Reasons To Accept:**

This is an interesting and meaningful work that presents a simple and effective method for generating reliable datasets. The datasets provided by the authors are of great help to the development of the NLP community.

**Reasons To Reject:**

The author uses GPT-4 to evaluate the data quality. I still feel that manual evaluation based on random sampling is necessary, because GPT-4 still faces the problem of model hallucinations.

**Reproducibility:**

4: Could mostly reproduce the results, but there may be some variation because of sample variance or minor variations in their interpretation of the protocol or method.

**Reviewer Confidence:**

4: Quite sure. I tried to check the important points carefully. It's unlikely, though conceivable, that I missed something that should affect my ratings.

---

> ### Author Rebuttal · Authors · 2023-08-28
>
> Thanks so much for your valuable comments!
>
> W1: GPT-4 is not reliable for evaluating the data quality and manual evaluation is needed.
>
> A1: Thanks for pointing this out. In the evaluation, we use the GPT-4 to compare if the assistant's responses are contradictory to the references, instead of letting GPT-4 judge the truthfulness of the responses with its own knowledge base.
>
> **We have supplemented the human evaluation following a similar pattern.** We randomly sample 50 English dialogues each for Alpaca, Baize, and RefGPT about factual knowledge. And 2 humans evaluate the truthfulness of the dialogues according to the references. The accuracies of Alpaca, Baize, and RefGPT are 54.0, 50.0, and 98.0. The results by humans have a similar trend as the GPT-4 evaluation (50.2, 47.2, 97.5).

---

### Official Review · Reviewer_7RQX · 2023-08-05

**Soundness:** 3

**Excitement:**

3: Ambivalent: It has merits (e.g., it reports state-of-the-art results, the idea is nice), but there are key weaknesses (e.g., it describes incremental work), and it can significantly benefit from another round of revision. However, I won't object to accepting it if my co-reviewers champion it.

**Paper Topic And Main Contributions:**

This paper introduces RefGPT, a method for generating dialogues using LLMs (e.g., GPT-4) without hallucinations, which are common in LLMs' outputs. To achieve that, the dialogues are generated from reference texts that can be trusted, such as Wikipedia articles. The method also involves an automatic construction of the dialogue template, including all the turns, and parameters for the individual turns to customize the conversation (e.g., the desired length of the response, tone used, etc.). The authors create two large datasets, with up to 100K dialogues each, using this method. In their automatic evaluation using GPT-4, they show the truthfulness of dialogues produced by their method is significantly higher than that of dialogues produced by other methods (97% vs. 50%).

**Questions For The Authors:**

- A.) Is the number of turns limited to 4? It appears so from Table 1. And are those 4 turns per user and assitant each, or 4 overall turns?
- B.) I found Section 5.1.1 very hard to understand the way it is currently written, especially the description of the evaluation setup for Self-Instruct and Baize. What do you mean by "we follow the Alpaca that we randomly select three {question, answer} pairs (from other references)..."? And what is a seed? What are the {question, answer} pairs for? They are nowhere to be seen in the examples in Table 3 and Appendix C. Are the {question, answer} pairs used instead of references when generating dialogues? I'm sorry, I just find it impossible to follow what is going on in this whole evaluation.
- C.) For the reference quality experiments in Section 5.2.1, did you try using some linguistic noise (e.g., replacing certain words with random words, shuffling sequences of words, removing negations, etc.) instead of adding random HTML labels? The model might simply ignore the HTML code for the most part, since it's typically not there in references.
- D.) From what noise level did the truthfulness become really bad, such as below 50%? If 20% of noise only reduced it from 98% to 96%, it makes me wonder if even 50% of noise would not have that big of an impact. Or otherwise I don't understand what the purpose of adding this type of noise really was in this experiment.

EDIT: Thank you for answering my questions and providing further clarifications. I have increased my scores accordingly.

**Reasons To Accept:**

- Emphasis on truthfulness of the generated dialogues and eliminating hallucination.
- Generating a whole dialogue in a single LLM query, which makes the dataset creation significantly faster (and presumably cheaper if using a paid API) than generating a dialogue turn by turn.
- Customizability of the dialogue templates for greater diversity of the dialogues produced using this method.
- Creation of two large multi-turn dialogue datasets that can be used to train perhaps much smaller models.

**Reasons To Reject:**

- The creation of the conversational style and dialogue content pools could require a substantial amount of manual work and possibly a certain amount of domain knowledge (depending on the task) in order for the pools to support diverse conversations when generating 100K of them. There are no details provided on the size of these pools and how they were constructed, so unfortunately it's hard to judge.
- Some sections are not written very clearly, hampering the understanding of the methods used and experiments performed (see an example in the questions section below).
- Most claims about truthfulness in Section 5 are based on automatic evaluations performed by GPT-4. How do we know we can trust GPT-4 for this task? At least a small study using human annotators would have been helpful to convince the audience of GPT-4's suitability for this type of evaluation.

**Reproducibility:**

4: Could mostly reproduce the results, but there may be some variation because of sample variance or minor variations in their interpretation of the protocol or method.

**Reviewer Confidence:**

3: Pretty sure, but there's a chance I missed something. Although I have a good feel for this area in general, I did not carefully check the paper's details, e.g., the math, experimental design, or novelty.

**Typos Grammar Style And Presentation Improvements:**

- L262: "ends" -> "end"
- L395: "can" -> "to"?
- L475: extra "are"

---

> ### Author Rebuttal · Authors · 2023-08-28
>
> Thanks for your valuable and insightful feedback!
>
> W1: The creation of conversational style and content pools could require a substantial amount of manual work and domain knowledge to generate 100k dialogues.
>
> A1: RefGPT can work well even without style and content pools. These additional settings only need a small amount of manual work for further customization. For example, in the generation of the RefGPT-Code dataset, the style pool, e.g. “asking like a young man”, has a size of about 10. And content pool, e.g. “writing、discussing the code”, has a size of about 20. We will supplement these details in our paper later.
>
> The domain knowledge is only needed for designing the task description in the dialogue setting and can be reused to generate a lot of dialogues. For example, in Figure 1, the code creation scenario has the task description: The user has a task related to the above code and wants to solve it with a computer program ... We generate 30k dialogues of code creation using only this description.
>
> **In RefGPT, the diversity of the dialogues mainly comes from the references rather than different designs in the content pool and style pool.**
>
>
>
> W2: Some sections are not written very clearly (see an example in the questions section).
>
> A2: See the explanation in Q2.
>
>
>
> W3: Truthfulness evaluation based on automatic evaluations performed by GPT-4 is not trustful. Using human annotators would have been helpful.
>
> A3: Thanks for pointing this out. In the evaluation, we use the GPT-4 to compare if the assistant's responses are contradictory to the references, instead of letting GPT-4 judge the truthfulness of the responses with its own knowledge base.
>
> **We have supplemented the human evaluation following a similar pattern.** We randomly sample 50 English dialogues each for Alpaca, Baize, and RefGPT about factual knowledge. And 2 humans evaluate the truthfulness of the dialogues according to the references. The accuracies of Alpaca, Baize, and RefGPT are 54.0, 50.0, and 98.0. The results by humans have a similar trend as the GPT-4 evaluation (50.2, 47.2, 97.5).
>
>
>
> W4: Reproductability: hard pressed to reproduce the results.
>
> A4: We have provided our prompts of RefGPT in Appendix A and the dataset examples in Appendix B. We will release our code and datasets later and they can easily be reproduced.
>
>
>
> Q1: Is the number of turns limited to 4? Are those 4 turns per user and assistant each, or 4 overall turns?
>
> A5: Yes, we limit the number of turns in the RefGPT datasets to 4. However, the RefGPT method can generate dialogues of more than 4 turns. And one turn contains one user question and one assistant response.
>
>
>
> Q2: The description of the evaluation setup for Self-Instruct (Alpaca) and Baize in Section 5.1.1 is hard to understand.
>
> A6: Alpaca uses three {instruction, output} pairs as the seeds to generate a new instruction, as we call {question, answer} pairs in the paper. Different from Alpaca using human-written instructions as the seed pool, we use the GPT-4 to generate instructions from selected references and constrain the answers (outputs) to be included in the references. The answers are double-checked by humans. The truthfulness evaluation process of Alpaca is as follows:
>
> ```
> #Seed Pool:
> {question1, answer1, reference1}, {question2, answer2, reference2}, ...
>
> #Generation:
> Input:
> {question1, answer1}
> {question2, answer2}
> {question3, answer3}
> question 4
> Output: the response to question 4. (response 4)
>
> #Evaluation:
> GPT-4/Human judges if response 4 to question 4 is correct (according to reference 4).
> ```
>
> Baize only uses the questions as seeds, which are generated in a similar pattern as Alpaca.
>
> We will add more examples to better clarify the evaluation process according to your advice.
>
>
>
> Q3: For the reference quality experiments in Section 5.2.1, did you try using some linguistic noise (e.g., replacing or shuffling words, removing negations, etc.) instead of adding random HTML labels?
>
> A7: We do not use linguistic noise because such noise changes the meaning of the references and leads to untruthful dialogues. We want the generated dialogues to strictly conform with the references instead of revising the references even if the references are incorrect. **The motivation for using HTML labels as noise is that many references may come from the crawled data on the websites and contain many HTML labels as noise if we do not clean the data carefully.** Due to the reliability of GPT-4, RefGPT can generate truthful dialogues despite the dirty reference data.
>
>
>
> Q4: From what noise level did the truthfulness become really bad, such as below 50%? I don't understand the purpose of adding this type of noise.
>
> A8: The motivation we add the HTML labels as noise can be checked in Q3A7 above. We choose 10% or 20% because many references crawled from websites may not be carefully cleaned and remain a few HTML labels. We also supplement more experiments on more noise levels shown below (0 ~ 20 results are different from the paper, please see Reviewer7XKU Q2).
>
> We can see GPT-4 can mostly generate truthful data from dirty references.
>
> | Noise Level | Truthfulness |
> | ----------- | ------------ |
> | 0           | 96.5         |
> | 10          | 96.2         |
> | 20          | 94.8         |
> | 30          | 93.3         |
> | 50          | 90.1         |
> | 70          | 86.2         |

---

### Meta-Review · Area_Chair_SVcD · 2023-09-19

**Recommendation:** 3

**Metareview:**

This paper introduces RefGPT, a method for generating truthful dialogue datasets using LLMs. Two reviewers pointed out that the truthfulness evaluation reported in the paper is based on automatic model-based evaluation with GPT-4, which cannot substitute for true human evaluation. The author response supplemented with additional human evaluation results and addressed the concern. One concern regarding the novelty -- improving LLM generation and reducing hallucination by providing additional textual information as conditional input is a common practice -- was not sufficiently addressed.

---

### Decision · Program_Chairs · 2023-10-07

**Decision:**

Accept-Findings

**Comment:**

This paper introduces RefGPT, a method for generating truthful dialogue datasets using LLMs. Two reviewers pointed out that the truthfulness evaluation reported in the paper is based on automatic model-based evaluation with GPT-4, which cannot substitute for true human evaluation. The author response supplemented with additional human evaluation results and addressed the concern. One concern regarding the novelty -- improving LLM generation and reducing hallucination by providing additional textual information as conditional input is a common practice -- was not sufficiently addressed.